# Scaling Off-Policy Reinforcement Learning with Batch and Weight Normalization

## Abstract

Reinforcement learning has achieved significant milestones, but sample efficiency remains a bottleneck for real-world applications. Recently, CrossQ has demonstrated state-of-the-art sample efficiency with a low update-to-data (UTD) ratio of 1. In this work, we explore CrossQ's scaling behavior with higher UTD ratios. We identify challenges in the training dynamics, which are emphasized by higher UTD ratios. To address these, we integrate weight normalization into the CrossQ framework, a solution that stabilizes training, has been shown to prevent potential loss of plasticity and keeps the effective learning rate constant. Our proposed approach reliably scales with increasing UTD ratios, achieving competitive performance across 25 challenging continuous control tasks on the DeepMind Control Suite and Myosuite benchmarks, notably the complex `dog` and `humanoid` environments. This work eliminates the need for drastic interventions, such as network resets, and offers a simple yet robust pathway for improving sample efficiency and scalability in model-free reinforcement learning.

## 1. Introduction

Reinforcement Learning (RL) has shown great successes in recent years, achieving breakthroughs in diverse areas. Despite these advancements, a fundamental challenge that remains in RL is enhancing the sample efficiency of algorithms. Indeed, in real-world applications, such as robotics, collecting large amounts of data can be time-consuming, costly, and sometimes impractical due to physical constraints or safety concerns. Thus, addressing this is crucial to make RL methods more accessible and scalable.

[1]Anonymous Institution, Anonymous City, Anonymous Region, Anonymous Country. Correspondence to: Anonymous Author <anon.email@domain.com>.

Preliminary work. Under review by the International Conference on Machine Learning (ICML). Do not distribute.

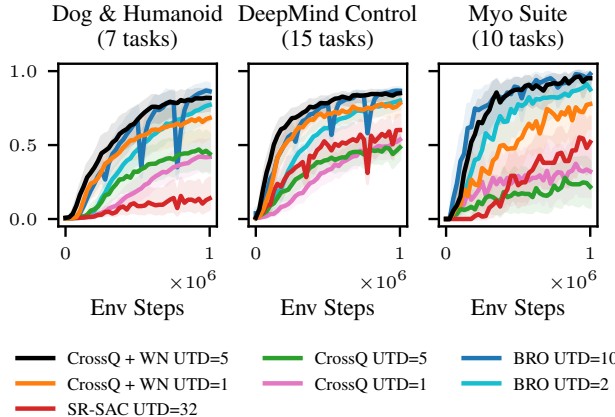

Figure 1. *CrossQ + WN UTD=5 is competitive to BRO UTD=10.* In comparison, our proposed CrossQ + WN is a simple algorithm that does not require extra exploration policies or full parameter resets. We present results for 25 complex continuous control tasks from the DMC and MyoSuite benchmarking suites. 1.0 marks the maximum score achievable on the respective benchmarks (DMC *return* up to 1000 / Myosuite up to 100% *success rate*).

Different approaches have been explored to address the problem of low sample efficiency in RL. Model-based RL, on the one hand, attempts to increase sample efficiency by learning dynamic models that reduce the need for collecting real data, a process often expensive and time-consuming (Sutton, 1990; Janner et al., 2019; Feinberg et al., 2018; Heess et al., 2015). Model-free RL approaches, on the other hand, have explored increasing the number of gradient updates on the available data, referred to as the update-to-data (UTD) ratio (Nikishin et al., 2022; D'Oro et al., 2022), modifying network architectures (Bhatt et al., 2024), or both (Chen et al., 2021; Hiraoka et al., 2021; Hussing et al., 2024; Nauman et al., 2024).

In this work, we build upon CrossQ (Bhatt et al., 2024), a recent model-free RL algorithm that recently showed state-of-the-art sample efficiency on the MuJoCo (Todorov et al., 2012) continuous control benchmarking tasks. Notably, the authors achieved this by carefully utilizing Batch Normalization (BN, Ioffe (2015)) within the actor-critic architecture.

A technique previously thought not to work in RL, as famously reported by Hiraoka et al. (2021) and others. The insight that Bhatt et al. (2024) offered is that one needs to carefully consider the different state-action distributions within the Bellman equation and handle them correctly to succeed. This novelty allowed CrossQ at a low UTD of 1 to outperform the then state-of-the-art algorithms that scaled their UTD ratios up to 20. Even though higher UTD ratios are more computationally expensive, they allow for larger policy improvements using the same amount of data.

This naturally raises the question: *How can we extend the sample efficiency benefits of CrossQ and* BN *to the high* UTD *training regime?* Which we address in this manuscript.

**Contributions.** In this work, we show that the vanilla CrossQ algorithm is brittle to tune on DeepMind Control (DMC) and Myosuite environments and can fail to scale reliably with increased compute. To address these limitations, we propose the addition of weight normalization (WN), which we show to be a simple yet effective enhancement that stabilizes CrossQ. We motivate the combined use of WN and BN on insights from the continual learning and loss of plasticity literature and connections to the effective learning rate. Our experiments show that incorporating WN not only improves the stability of CrossQ but also allows us to scale its UTD, thereby significantly enhancing sample efficiency.

## 2. Preliminaries

This section briefly outlines the required background knowledge for this paper.

**Reinforcement learning.** A Markov Decision Process (MDP) (Puterman, 2014) is a tuple $\mathcal{M} = \langle \mathcal{S}, \mathcal{A}, \mathcal{P}, r, \mu_0, \gamma \rangle$, with state space $\mathcal{S} \subseteq \mathbb{R}^n$, action space $\mathcal{A} \subseteq \mathbb{R}^m$, transition probability $\mathcal{P} : \mathcal{S} \times \mathcal{A} \to \Delta(\mathcal{S})$, the reward function $r : \mathcal{S} \times \mathcal{A} \to \mathbb{R}$, initial state distribution $\mu_0$ and discount factor $\gamma$. We define the RL problem according to Sutton & Barto (2018). A policy $\pi : \mathcal{S} \to \Delta(\mathcal{A})$ is a behavior plan, which maps a state $s$ to a distribution over actions $a$. The discounted cumulative return is defined as

$$\mathcal{R}(s, a) = \sum_{t=0}^{\infty} \gamma^t r(s_t, a_t),$$

where $s_0 = s$ and $a_0 = a$. Further, $s_{t+1} \sim \mathcal{P}(\,\cdot\,|s_t, a_t)$ and $a_t \sim \pi(\,\cdot\,|s_t)$. The Q-function of a policy $\pi$ is the expected discounted return $Q^\pi(s, a) = \mathbb{E}_{\pi, \mathcal{P}}[\mathcal{R}(s, a)]$.

The goal of an RL agent is to find an optimal policy $\pi^*$ that maximizes the expected return from the initial state distribution

$$\pi^* = \arg\max_\pi \mathbb{E}_{s \sim \mu_0}[Q^\pi(s, a)].$$

Soft Actor-Critic (SAC, Haarnoja et al. (2018)) addresses this optimization problem by jointly learning neural network representations for the Q-function and the policy. The policy network is optimized to maximize the Q-values, while the Q-function is optimized to minimize the Bellmann error

$$\mathcal{L} = \mathbb{E}_{\mathcal{D}}\left[\frac{1}{2}\left(Q_\theta(s_t, a_t) - \left(r(s_t, a_t) + \gamma \mathbb{E}_{\mathcal{P}}[V(s_{t+1})]\right)\right)^2\right],$$

where the value function is computed by taking an expectation over the learned Q function

$$V(s_{t+1}) = \mathbb{E}_{\mathcal{P}, \pi_\theta}[Q_{\bar\theta}(s_{t+1}, a_{t+1})]. \tag{1}$$

To stabilize the Q-function learning, Haarnoja et al. (2018) found it necessary to use a target Q-network in the computation of the value function instead of the regular Q-network. The target Q-network is structurally equal to the regular Q-network, and its parameters $\bar\theta$ are obtained via Polyak Averaging over the learned parameters $\theta$. While this scheme ensures stability during training by explicitly delaying value function updates, it also arguably slows down online learning (Plappert et al., 2018; Kim et al., 2019; Morales, 2020).

Instead of relying on target networks, CrossQ (Bhatt et al., 2024) addresses training stability issues by introducing Batch Normalization (BN, Ioffe (2015)) in its Q-function and achieves substantial improvements in sample and computational efficiency over SAC. A central challenge when using BN in Q networks is distribution mismatch: during training, the Q-function is optimized with samples $s_t, a_t$ from the replay buffer. However, when the Q-function is evaluated to compute the target values (Equation (1)), it receives actions sampled from the current policy $a_{t+1} \sim \pi_\theta(\,\cdot\,|s_{t+1})$. Those samples have no guarantee of lying within the training distribution of the Q-function. BN is known to struggle with out-of-distribution samples, as such, training can become unstable if the distribution mismatch is not correctly accounted for (Bhatt et al., 2024). To deal with this issue, CrossQ removes the separate target Q-function and evaluates both Q values during the critic update in a single forward pass, which causes the BN layers to compute shared statistics over the samples from the replay buffer and the current policy. This scheme effectively tackles distribution mismatch problems, ensuring that all inputs and intermediate activations are effectively forced to lie within the training distribution.

**Normalization techniques in RL.** Normalization techniques are widely recognized for improving the training of neural networks, as they generally accelerate training and improve generalization (Huang et al., 2023). There are many ways of introducing different types of normalizations into the RL framework. Most commonly, authors have used Layer Normalization (LN) within the network architectures to stabilize training (Hiraoka et al., 2021; Lyle et al., 2024). Recently, CrossQ has been the first algorithm to successfully use BN layers in RL (Bhatt et al., 2024). The addition of BN leads to substantial gains in sample efficiency. In contrast to

LN, however, one needs to carefully consider the different state-action distributions within the critic loss when integrating BN. In a different line of work, Hussing et al. (2024) proposed the integration of unit ball normalization and projected the output features of the penultimate layer onto the unit ball in order to reduce Q-function overestimation.

**Increasing update-to-data ratios.** Although scaling up the UTD ratio is an intuitive approach to increase the sample efficiency, in practice, it comes with several challenges. Nikishin et al. (2022) demonstrated that overfitting on early training data can inhibit the agent from learning anything later in the training. The authors dub this phenomenon the primacy bias. To address the primacy bias, they suggest to periodically reset the network parameters while retraining the replay buffer. Many works that followed have adapted this intervention (D'Oro et al., 2022; Nauman et al., 2024). While often effective, regularly resetting is a very drastic intervention and by design induces regular drops in performance. Since the agent has to start learning from scratch repeatedly, it is also not very computing efficient. Finally, the exact reasons why parameter resets work well in practice are not yet well understood (Li et al., 2023). Instead of resetting there have also been other types of regularization that allowed practitioners to train stably with high UTD ratios. Janner et al. (2019) generate additional modeled data, by virtually increasing the UTD. In REDQ, Chen et al. (2021) leverage ensembles of Q-functions, while Hiraoka et al. (2021) use dropout and LN to effectively scale to higher UTD ratios.

## 3. CrossQ fails to scale up stably

Bhatt et al. (2024) demonstrated CrossQ's state-of-the-art sample efficiency on the MuJoCo task suite (Todorov et al., 2012), while at the same time also being very computationally efficient. However, on the more extensive DMC and Myosuite task suites, we find that CrossQ requires tuning. We further find that it works on some, but not all, environments stably and reliably.

**Mixed performance of CrossQ.** Figure 2 shows CrossQ training performance on a subset of DMC tasks. Namely, the dog-stand, dog-trot and humanoid-walk, selected for their varying difficulty levels to demonstrate a wide range of behaviors, including both successful learning and challenges encountered during training. The figure compares a SAC baseline with standard hyperparameters against tuned CrossQ agents with UTD ratios of 1 and 5, where the hyperparameters were identified through a grid search over learning rates and network sizes, as detailed in Table 1. The first row of the figure shows the IQM training performance and 95% confidence intervals for each agent across 10 seeds. Here, we identify three different train-

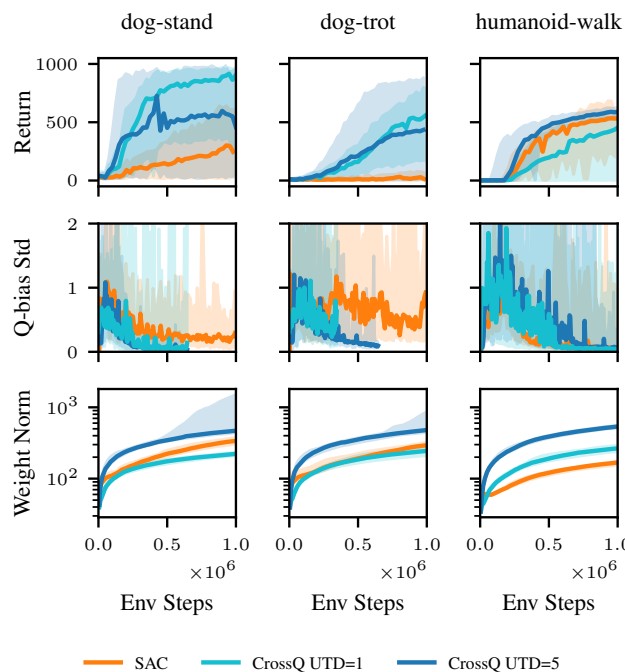

Figure 2. *Q-bias and weight norms.* CrossQ critic weight norms increase significantly with increasing UTD ratios.

ing behaviors. On dog-stand CrossQ trains stably at UTD= 1, but increasing the UTD to 5 introduces instabilities and decreases performance. On dog-trot, both UTD ratios perform very similarly. Finally, on humanoid-walk, UTD=5 outperforms UTD=1, although it merely manages to catch up to the SAC baseline in this case. Overall, for all CrossQ runs we notice very large confidence intervals.

**Q-function bias analysis.** The second row in Figure 2 shows the standard deviation of the normalized Q-function bias. This bias measures how much the Q-function is overestimating or underestimating the true expected return of the current policy.

To compute the normalized Q-function bias, we follow the protocol of Chen et al. (2021). We gather 5 trajectories in the environment using the current policy and use each trajectory's first 350 state-action pairs to calculate the bias. For this, we compare the cumulative discounted rewards for each state-action pair with its Q-value predicted by the Q-function. This bias is then normalized using the cumulative discounted rewards. The mean over these normalized Q-function biases measures the expected bias. Even if the mean is high, as long as the bias is consistent, the selected actions of the policy do not change and, therefore, it is not a problem (Van Hasselt et al., 2016). If the standard deviation is high, the change in bias is high, which might hinder learning. Thus, following the work of Chen et al. (2021);

Bhatt et al. (2024), we focus on the standard deviation.

We find large fluctuations for the Q-function bias standard deviation with all three agents across environments. And even on dog-stand, where CrossQ UTD=5 does not learn reliably, it maintains a small Q-function bias standard deviation. From these mixed results, we conclude that we cannot directly link the standard deviation of the Q-function bias to the learning performance. While this does not mean that for larger UTD ratios, the Q-bias would not explode. Rather, it means that, in our case, the Q-bias is not at fault, and the root cause for unstable training lies elsewhere.

**Growing network parameter norms.** The third row of Figure 2 displays the sum over the L2 norms of the dense layers in the critic network. This includes three dense layers, each with a hidden dimension of $512$.

All three baselines exhibit growing network weights over the course of training. We find that the effect is particularly pronounced for CrossQ with increasing UTD ratios. We further find that in the second training phase, the spread of network weight norms increases for the dog runs with large confidence intervals. This is visualized by the large spread of the shaded areas, which show the $95\%$ inter percentile ranges for the weight norms.

Growing network weights have been linked to a loss of plasticity, a phenomenon where networks become increasingly resistant to parameter update, which can lead to premature convergence (Elsayed et al., 2024). Additionally, the growing magnitudes pose a challenge for optimization, connected to the issue of growing activations, which has recently been analyzed by Hussing et al. (2024). Further, growing network weights decrease the effective learning rate when the networks contain normalization layers (Van Hasselt et al., 2019; Lyle et al., 2024).

In summary, the scaling results for vanilla CrossQ are mixed. While increasing UTD ratios is known to yield increased sample efficiency, if careful regularization is used (Janner et al., 2019; Chen et al., 2021; Nikishin et al., 2022), CrossQ alone with BN cannot benefit from it. We notice that with increasing UTD ratios, CrossQ's weight layer norms grow significantly faster and overall larger. This observation motivates us to further study the weight norms in CrossQ to increase UTD ratios.

## 4. Combining batch normalization and weight normalization for scaling up

Inspired by the combined insights of Van Hasselt et al. (2019) and Lyle et al. (2024), we propose to integrate CrossQ with Weight Normalization (WN) as a means of counteracting the rapid growth of weight norms we observe with increasing update-to-data (UTD) ratios.

Our approach is based on the following reasoning: Due to the use of BN in CrossQ, the critic network exhibits scale invariance, as previously noted by Van Laarhoven (2017).

**Theorem 4.1** (Van Laarhoven (2017))**.** *Let* $f(\boldsymbol{X}; \boldsymbol{w}, b, \gamma, \beta)$ *be a function, with inputs* $\boldsymbol{X}$ *and parameters* $\boldsymbol{w}$ *and* $\boldsymbol{b}$ *and* $\gamma$ *and* $\beta$ *batch normalization parameters. When* $f$ *is normalized with batch normalization,* $f$ *becomes scale-invariant with respect to its parameters, i.e.,*

$$f(\boldsymbol{X}; c\boldsymbol{w}, cb, \gamma, \beta) = f(\boldsymbol{X}; \boldsymbol{w}, b, \gamma, \beta),$$

*with scaling factor* $c > 0$.

*Proof. Appendix A*

This property allows us to introduce WN as a mechanism to regulate the growth of weight norms in CrossQ without affecting the critic's outputs. Further, it can be shown, that for such a scale invariant function, the gradient scales inversely proportionally to the scaling factor $c > 0$.

**Theorem 4.2** (Van Laarhoven (2017))**.** *Let* $f(\boldsymbol{X}; \boldsymbol{w}, b, \gamma, \beta)$ *be a scale-invariant function. Then, the gradients of* $f$ *scale inversely proportional to the scaling factor* $c \in \mathbb{R}$ *of its parameters* $\boldsymbol{w}$,

$$\nabla f(\boldsymbol{X}; c\boldsymbol{w}, cb, \gamma, \beta) = \nabla f(\boldsymbol{X}; \boldsymbol{w}, b, \gamma, \beta)/c.$$

*Proof. Appendix B*

Recently, Lyle et al. (2024) demonstrated that the combination of LN and WN can help mitigate loss of plasticity. Since the gradient scale is inversely proportional to $c$, keeping norms constant helps to maintain a stable effective learning rate (ELR, Van Hasselt et al. (2019)), further enhancing training stability.

We conjecture that maintaining a stable ELR could also be beneficial when increasing the UTD ratios in continuous control RL. As the UTD ratio increases, the networks are updated more frequently with each environment interaction. Empirically, we find that the network norms tend to grow quicker with increased UTD ratios (Figure 2), which in turn decreases the ELR even quicker and could be the case for instabilities and low training performance. As such, we empirically investigate the effectiveness of combining CrossQ with WN with increasing UTD ratios.

**Implementation details.** We apply WN to the first two linear layers, ensuring that their weights remain unit norm after each gradient step by projecting them onto the unit ball, similar to Lyle et al. (2024). Since this constraint effectively stabilizes the ELR in these layers, we found it beneficial to slightly reduce the learning rate from $1e-3$ to $1e-4$. While we could employ a learning rate schedule (Lyle et al., 2024) we did not investigate this here. Additionally, we impose

weight decay on all parameters that remain unbounded—specifically the final dense output layer. In practice, we use AdamW (Loshchilov, 2017) with a decay of 0 (which falls back to vanilla Adam (Kingma, 2014)) for the normalized intermediate dense layers and $1e-2$ otherwise. We chose a maximum UTD of 5 for our experiments due to our computational budget and good strong in sample efficiency.

**Target networks.** CrossQ famously removed the target networks from the actor-critic framework and showed that using BN training remains stable even without them (Bhatt et al., 2024). While we find this to be true in many cases, we find that especially in DMC, the re-integration of target networks can help stabilize training overall (see Section 5.4). However, not surprisingly, we find that the integration of target networks with BN requires careful consideration of the different state-action distributions between the $s, a$ and $s', a' \sim \pi(s')$ exactly as proposed by Bhatt et al. (2024). To satisfy this, we keep the joined forward pass through both the critic network as well as the target critic network. We evaluate both networks in *training mode*, i.e., they calculate the joined state-action batch statistics on the current batches. As is common, we use Polyak-averaging with a $\tau = 0.005$ from the critic network to the target network.

# 5. Experiments

To evaluate the effectiveness of our proposed CrossQ + WN method, we conduct a comprehensive set of experiments on the DeepMind Control Suite (Tassa et al., 2018) and MyoSuite (Caggiano et al., 2022) benchmarks. Our primary goal is to investigate the scalability of CrossQ + WN with increasing UTD ratios and to assess the stabilizing effects of combining CrossQ with WN. We compare our approach to several baselines, including the recent BRO (Nauman et al., 2024), CrossQ (Bhatt et al., 2024) and SR-SAC (D'Oro et al., 2022) a version of SAC (Haarnoja et al., 2018) with high UTD ratios and network resets.

## 5.1. Experimental setup

Our implementation is based on the SAC implementation of `jaxrl` codebase (Kostrikov, 2021). We implement CrossQ following the author's original codebase and add the architectural modifications introduced by (Bhatt et al., 2024), incorporating batch normalization in the actor and critic networks. We extend this approach by introducing WN to regulate the growth of weight norms and prevent loss of plasticity and add target networks. We perform a grid search to focus on learning rate selection and layer width.

We evaluate 25 diverse continuous control tasks, 15 from DMC and 10 from MyoSuite. These tasks vary significantly in complexity, requiring different levels of fine motor control and policy adaptation with high dimensional state spaces up to $\mathbb{R}^{223}$.

Each experiment is run for 1 million environment steps and across 10 random seeds to ensure statistical robustness. We evaluate agents every $25,000$ environment steps for 5 trajectories. As proposed by Agarwal et al. (2021), we report the interquartile mean (IQM) and $95\%$ stratified bootstrap confidence intervals (CIs) of the return, if not otherwise stated.

For the BRO baseline results, for computational reasons, we take the official evaluation data the authors provide. The official BRO codebase is also based on `jaxrl`, and the authors followed the same evaluation protocol, making it a fair comparison.

## 5.2. Weight normalization allows CrossQ to scale effectively

We provide empirical evidence for our hypothesis that controlling the weight norm and, thereby, the ELR can stabilize training. We show that through the addition of WN, CrossQ + WN shows stable training and can stably scale with increasing UTD ratios.

Figure 3 shows per environment results of our experiments encompassing all 25 tasks evaluated across 10 seeds each. Based on that, Figure 1 shows aggregated performance over all environments from Figure 3 per task suite, with a separate aggregation for the most complex `dog` and `humanoid` environments.

These results show that CrossQ + WN UTD=5 is competitive to the BRO baseline on both DMC and Myosuite, especially on the more complex `dog` and `humanoid` tasks. Notably, CrossQ + WN UTD=5 uses only half the UTD of BRO and does not require any parameter resets and no additional exploration policy. Further, it uses $\sim 90\%$ fewer network parameters—BRO reports $\sim 5M$, while our proposed CrossQ + WN uses only $\sim 600k$ (these numbers vary slightly per environment, depending on the state and action dimensionalities).

In contrast, vanilla CrossQ UTD=1 exhibits much slower learning on most tasks and, in some environments, fails to learn performant policies. Moreover, the instability of vanilla CrossQ at UTD=5 is particularly notable, as it does not reliably converge across environments.

These findings highlight the necessity of incorporating additional normalization techniques to sustain effective training at higher UTD ratios. This leads us to conclude that CrossQ benefits from the addition of WN, which results in stable training and scales well with higher UTD ratios. The resulting algorithm can match or outperform state-of-the-art baselines on the continuous control DMC and Myosuite benchmarks while being much simpler algorithmically.

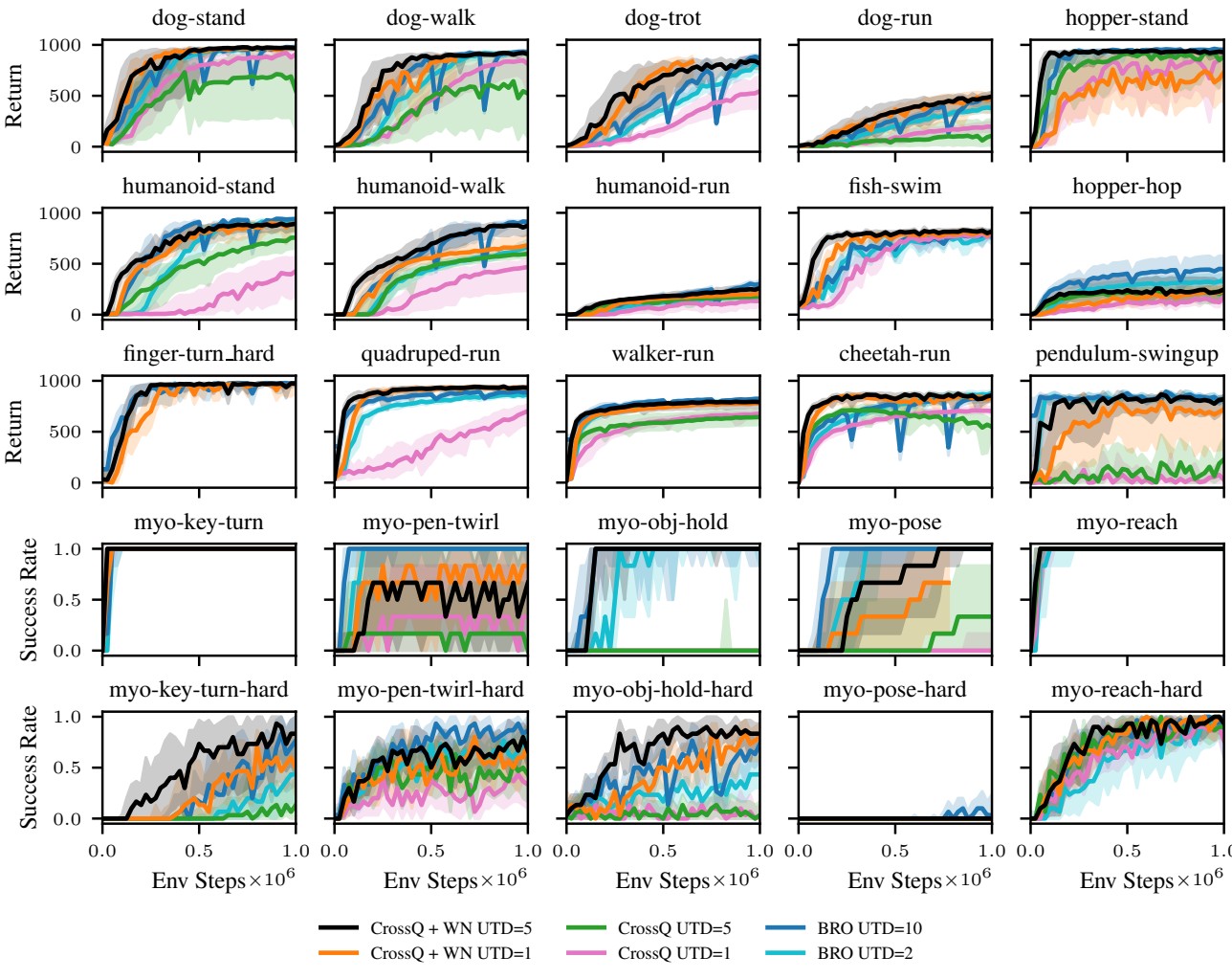

*Figure 3. CrossQ* WN + UTD=5 *against baselines.* We compare our proposed CrossQ + WN UTD=5 against two baselines, BRO (Nauman et al., 2024) and SR-SAC UTD=32. Results are reported on all 15 DMC and 10 Myosuite tasks. We plot the IQM and 95% CIs over 10 random random seeds. Our proposed approach proves competitive to BRO and outperforms the CrossQ baseline. We want to note that our approach achieves this performance without requiring any parameter resetting or additional exploration policies.

## 5.3. Stable scaling of CrossQ + WN with UTD ratios

To visualize the stable scaling behavior of CrossQ + WN we ablate across three different UTD ratios $\in \{1, 2, 5\}$. Figure 4 shows training curves aggregated over all 15 DMC tasks. As expected, CrossQ + WN shows reliable scaling behavior, with the learning curves ordered in increasing order accordance to their respective UTD ratio.

## 5.4. Hyperparameter ablation studies

We also ablate the different hyperparameters of CrossQ + WN UTD=5, by changing each one at a time. Figure 5 shows aggregated results of the final performances of each ablation. We will briefly discuss each ablation individually.

**Removing weight normalization.** Not performing weight normalization results in the biggest drop in performance across all our ablations. This loss is most drastic on the Myosuite tasks and often results in no meaningful learning. Showing that, as hypothesized, the inclusion of WN into the CrossQ framework yields great improvements in terms of sample efficiency and training stability, especially for larger UTD ratios.

**Update-to-data ratio.** As expected, decreasing the UTD ratio decreases the performance of CrossQ + WN, as demonstrated in the previous section. Aggregated over all DMC environments, this effect is the smallest, since this aggregation includes easier environments as well. Looking at the harder `dog` and `humanoid` tasks, as well as Myosuite,

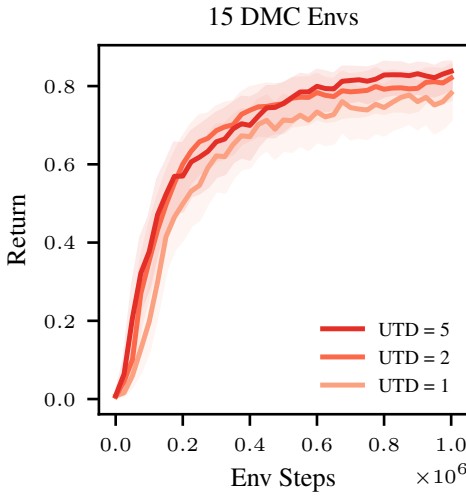

Figure 4. *CrossQ WN UTD scaling behavior.* We plot the IQM return and 95% confidence intervals for different UTD ratios ∈ {1, 2, 5}. The results are aggregated over 15 DMC environments and 10 random seeds each according to Agarwal et al. (2021). The sample efficiency scales reliably with increasing UTD ratios.

the effect is more pronounced. However, lower UTDs are already reasonably competitive in overall performance.

**Target networks.** Ablating the target networks shows that on Myosuite, there is no significant difference between using a target network and or no target network. Results on DMC differ significantly. There, removing target networks leads to a significant drop in performance, nearly as large as removing weight normalization. This finding is interesting, as it suggests that CrossQ + WN without target networks is not inherently unstable. But there are situations where the inclusion of target networks is required. Further investigating the role and necessity of target networks in RL is an interesting direction for future research.

## 6. Related work

RL has demonstrated remarkable success across various domains, yet sample efficiency remains a significant challenge, especially in real-world applications where data collection is expensive or impractical. Various approaches have been explored to address this issue, including model-based RL, UTD ratio scaling, and architectural modifications.

**Model-based RL** methods enhance sample efficiency by constructing predictive models of the environment to reduce reliance on real data collection (Sutton, 1990; Janner et al., 2019; Feinberg et al., 2018; Heess et al., 2015). However, such methods introduce additional complexity, computational overhead, and potential biases due to model inaccuracies.

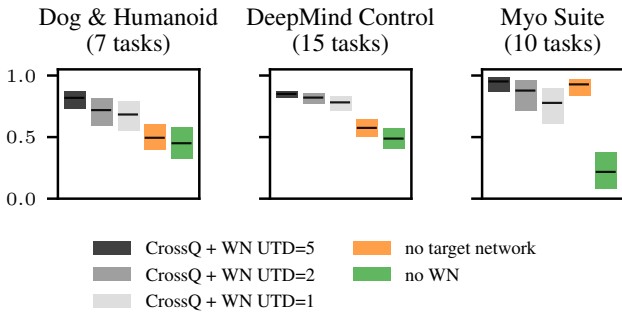

Figure 5. *Hyperparameter ablations.* We ablate CrossQ + WN with respect to the WN, target networks and different UTD.

**Update-to-data ratio scaling.** Model-free RL methods, including those utilizing higher UTD ratios, focus on increasing the number of gradient updates per collected sample to maximize learning from available data. High UTD training introduces several challenges, such as overfitting to early training data, a phenomenon known as primacy bias (Nikishin et al., 2022). This can be counteracted by periodically resetting the network parameters (Nikishin et al., 2022; D'Oro et al., 2022). However, network resets introduce abrupt performance drops. Alternative approaches use techniques such as Q-function ensembles (Chen et al., 2021; Hiraoka et al., 2021) and architectural changes (Nauman et al., 2024).

**Normalization techniques in RL.** Normalization techniques have long been recognized for their impact on neural network training. LN (Ba et al., 2016) and other architectural modifications have been used to stabilize learning in RL (Hiraoka et al., 2021; Nauman et al., 2024). Yet BN has only recently been successfully applied in this context (Bhatt et al., 2024), challenging previous findings, where BN in critics caused training to fail (Hiraoka et al., 2021). WN has been shown to keep ELRs stable and prevent loss of plasticity (Lyle et al., 2024), when combined with LN, making it a promising candidate for integration into existing RL frameworks.

## 7. Limitations & future work

In this work, we only consider continuous state-action benchmarking tasks. While our proposed CrossQ + WN performs competitively on these tasks, its performance on discrete state-action spaces or vision-based tasks remains unexplored. We plan to investigate this in future work.

## 8. Conclusion

In this work, we have addressed the instability and scalability limitations of CrossQ in RL by integrating WN. Our empirical results demonstrate that WN effectively stabilizes training and allows CrossQ to scale reliably with higher UTD ratios. The proposed CrossQ + WN approach achieves competitive or superior performance compared to state-of-the-art baselines across a diverse set of 25 complex continuous control tasks from the DMC and Myosuite benchmarks. These tasks include complex and high-dimensional humanoid and dog environments. This extension preserves simplicity while enhancing robustness and scalability by eliminating the need for drastic interventions such as network resets.

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

## A. Proof Scale Invariance

Proof of Theorem 4.1.

$$f(\boldsymbol{X}; c\boldsymbol{w}, cb, \gamma, \beta) = \frac{g(c\boldsymbol{X}\boldsymbol{w} + cb) - \mu(g(c\boldsymbol{X}\boldsymbol{w} + cb))}{\sigma(g(c\boldsymbol{X}\boldsymbol{w} + cb))}\gamma + \beta$$

$$= \frac{cg(\boldsymbol{X}\boldsymbol{w} + b) - c\mu(g(\boldsymbol{X}\boldsymbol{w} + b))}{|c|\sigma(g(\boldsymbol{X}\boldsymbol{w} + b))}\gamma + \beta$$

$$= \frac{g(\boldsymbol{X}\boldsymbol{w} + b) - \mu(g(\boldsymbol{X}\boldsymbol{w} + b))}{\sigma(g(\boldsymbol{X}\boldsymbol{w} + b))}\gamma + \beta = f(\boldsymbol{X}; \boldsymbol{w}, b, \gamma, \beta)$$

## B. Proof Inverse Proportional Gradients

To show that the gradients scale inversely proportional to the parameter norm, we can first write

$$f(\boldsymbol{X}; c\boldsymbol{w}, cb, \gamma, \beta) = \frac{g(c\boldsymbol{X}\boldsymbol{w} + cb) - \mu(g(c\boldsymbol{X}\boldsymbol{w} + cb))}{\sigma(g(c\boldsymbol{X}\boldsymbol{w} + cb))}\gamma + \beta$$

$$= \frac{g(c\boldsymbol{X}\boldsymbol{w} + cb)}{\sigma(g(c\boldsymbol{X}\boldsymbol{w} + cb))}\gamma - \frac{\mu(g(c\boldsymbol{X}\boldsymbol{w} + cb))}{\sigma(g(c\boldsymbol{X}\boldsymbol{w} + cb))}\gamma + \beta.$$

As the gradient of the weights is not backpropagated through the mean and standard deviation, we have

$$\nabla_w f(\boldsymbol{X}; c\boldsymbol{w}, cb, \gamma, \beta) = \frac{g'(c\boldsymbol{X}\boldsymbol{w} + cb)X}{|c|\sigma(g(\boldsymbol{X}\boldsymbol{w} + b))}\gamma.$$

The gradient of the bias can be computed analogously

$$\nabla_b f(\boldsymbol{X}; c\boldsymbol{w}, cb, \gamma, \beta) = \frac{g'(c\boldsymbol{X}\boldsymbol{w} + cb)}{|c|\sigma(g(\boldsymbol{X}\boldsymbol{w} + b))}\gamma.$$

# C. Hyperparameters

Table 1 gives an overview of the hyperparameters that were used for each algorithm that was considered in this work.

*Table 1.* Hyperparameters

| Hyperparameter | CrossQ | CrossQ + WN | SAC | SR-SAC | BRO |
|---|---|---|---|---|---|
| Critic learning rate | 0.0001 | 0.0001 | 0.0003 | 0.0003 | 0.0003 |
| Critic hidden dim | 512 | 512 | 256 | 256 | 512 |
| Actor learning rate | 0.0001 | 0.0001 | 0.0003 | 0.0003 | 0.0003 |
| Actor hidden dim | 256 | 256 | 256 | 256 | 256 |
| Initial temperature | 1.0 | 1.0 | 1.0 | 1.0 | 1.0 |
| Temperature learning rate | 0.0001 | 0.0001 | 0.0003 | 0.0003 | 0.0003 |
| Target entropy | $|A|$ | $|A|$ | $|A|$ | $|A|$ | $|A|$ |
| Target network momentum | 0.005 | 0.005 | 0.005 | 0.005 | 0.005 |
| UTD | 1,5 | 1,5 | 1 | 32 | 10 |
| Number of critics | 2 | 2 | 2 | 2 | 1 |
| Action repeat | 2 | 2 | 2 | 2 | 1 |
| Discount | 0.99 (DMC) 0.95 (Myo) | 0.99 (DMC) 0.95 (Myo) | 0.99 (DMC) 0.95 (Myo) | 0.99 (DMC) 0.95 (Myo) | 0.99 (DMC) 0.99 (Myo) |
| Optimizer | Adam | AdamW | Adam | Adam | AdamW |
| Optimizer momentum $(\beta_1, \beta_2)$ | (0.9, 0.999) | (0.9, 0.999) | (0.9, 0.999) | (0.9, 0.999) | (0.9, 0.999) |
| Policy delay | 3 | 3 | 1 | 1 | 1 |
| Warmup transitions | 5000 | 5000 | 10000 | 10000 | 10000 |
| AdamW weight decay critic | 0.0 | 0.01 | 0.0 | 0.0 | 0.0001 |
| AdamW weight decay actor | 0.0 | 0.01 | 0.0 | 0.0 | 0.0001 |
| AdamW weight decay temperature | 0.0 | 0.0 | 0.0 | 0.0 | 0.0 |
| Batch Normalization momentum | 0.99 | 0.99 | N/A | N/A | N/A |
| Reset Interval of networks | N/A | N/A | N/A | every 80k steps | *at 15k, 50k, 250k, 500k and 750k steps* |
| Batch Size | 256 | 256 | 256 | 256 | 128 |

# D. Individual training curves for ablation results

Here, we provide detailed individual training curves for each ablation experiment conducted in our study. Figure 6 shows experiments with no WN, no target network, CrossQ with WN and UTD=1, and CrossQ with WN and UTD=5.

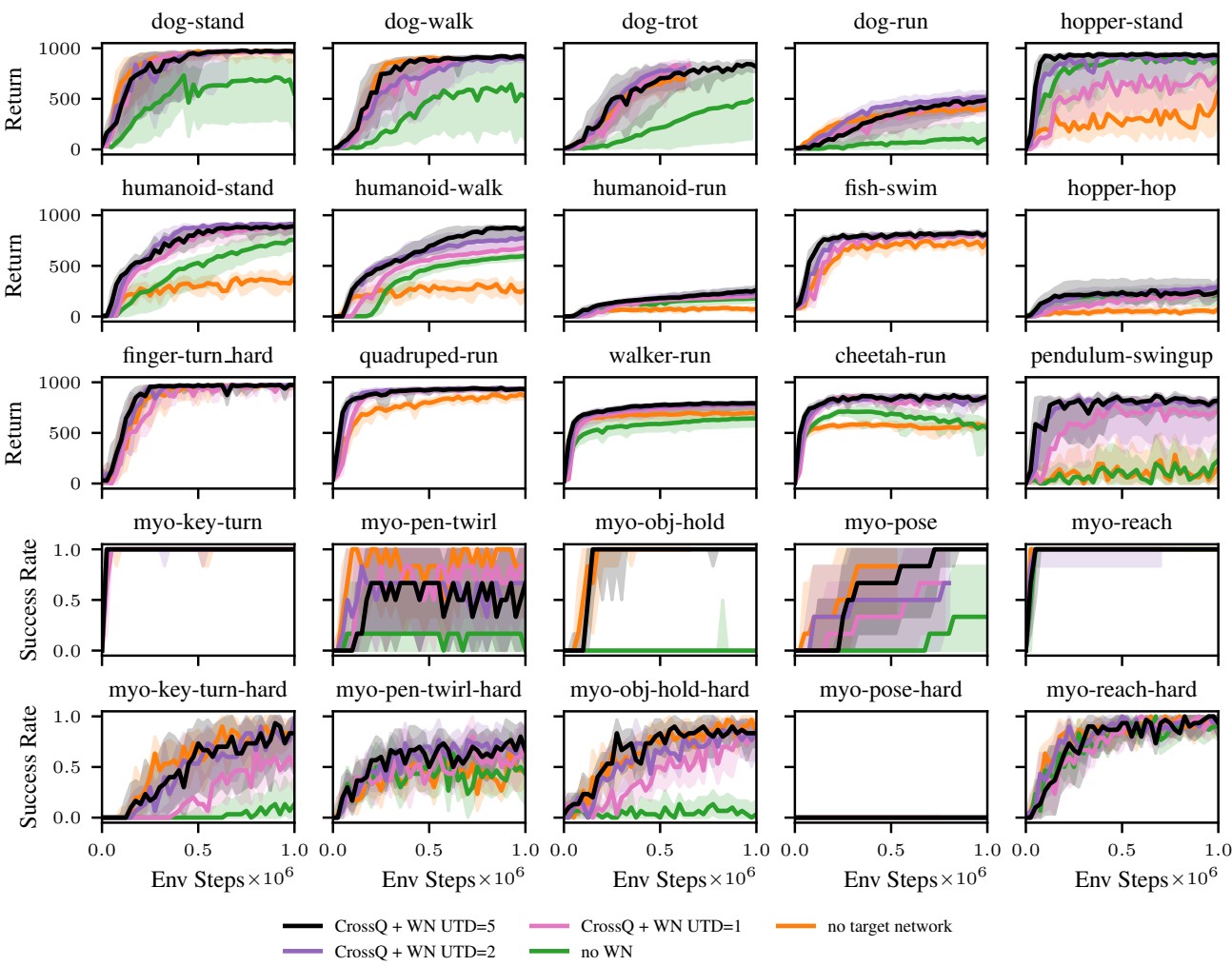

Figure 6. Individual training curves for ablations

