# OpenReview forum: "Scaling Off-Policy Reinforcement Learning with Batch and Weight Normalization"
_ICML.cc/2025/Conference — Submitted to ICML 2025_

### Official Review · Reviewer_9s2q · 2025-03-10

**Overall Recommendation:** 4

**Summary:**

The paper discusses the failure of a previously proposed DRL algorithm, CrossQ, to reliably scale up to more complex environments than those considered in the original paper. To achieve this, the authors propose to use weight normalization.



### Update after rebuttal period

With additional context, I support the acceptance of this paper. All major questions were adequately addressed.

**Claims And Evidence:**

The claims made in the paper are backed up with a standard set of experiments on DMC and MyoSuite (two environments where CrossQ previously failed to achieve reliable returns). As the main goal of the paper was to solve this issue, the evidence matches the claims.

**Essential References Not Discussed:**

Note that MAD-TD is a very recent paper, and so I do not consider it an essential reference and do not account for it in my score. However, it influenced the questions I asked so I am listing it here.

[1] MAD-TD: MODEL-AUGMENTED DATA STABILIZES HIGH UPDATE RATIO RL, Voelcker, C. et al., ICLR 2025

[2] TD-MPC2: Scalable, Robust World Models for Continuous Control, Hansen, N., ICLR 2024

**Experimental Designs Or Analyses:**

The experimental design is standard. As a minor nitpick, BRO and CrossQ are evaluated with differing action repeat values, which has recently been shown to impact the performance of model-free methods massively [1]. It would be good to account for this design choice. ("The official BRO codebase is also based on jaxrl, and the authors followed the same evaluation protocol, making it a fair comparison." is therefore also technically an incorrect claim)

For easier comparison, it would be nice to have the plots in Figure 5 include the baselines.

**Methods And Evaluation Criteria:**

The chosen method and evaluation is standard and applicable.

**Other Comments Or Suggestions:**

n/a

**Other Strengths And Weaknesses:**

Overall the paper is very readable, kudos!

**Questions For Authors:**

The premise creates several questions which I would encourage the authors to discuss in more detail:

1) Why was WeightNorm specifically chosen to improve CrossQ? Could other proposals, such as Unit Sphere Normalization which was referenced in the paper also work?

2) What is the relative relevance of the CrossQ part of the architecture? Is weightnorm alone sufficient for strong performance, and if not, why not?

3) In an additional vein, why is BatchNorm useful for avoiding the reset strategy employed by BRO? Do additional resets give CrossQ any benefits? Other recent works, such as [1] (see below) have argued that deteriorating performance corresponds to out-of-distribution actions in off-policy learning, would such a hypothesis fit with the evidence presented here?

4) How does CrossQ perform compared to model-based approaches such as [1] and [2]?

**Relation To Broader Scientific Literature:**

The paper is somewhat incremental, but still has some value for the  community. It essentially combines the insights from CrossQ with the current literature which focuses on the impact of LayerNorm and other approaches.

**Theoretical Claims:**

No new theoretical claims are presented. Theorems used in the work are from prior literature. Overall, I would encourage the authors to connect the presented theory more closely with the results in the paper. As it stands, I am not fully sure how scale invariance helps the architecture.

---

> ### Author Rebuttal · Authors · 2025-04-01
>
> We thank the reviewer for their review and positive feedback, as well as the additional questions. We will extend the paper with discussions, additional experiments for each question, and mentioned baselines and feedback. To answer them here:
>
> - **Could Unit Sphere Normalization work (instead of the proposed WN)?** We need to differentiate between normalizing the features and normalizing the weights. The proposed Unit Sphere Normalization in Hussing et. al 2024, is with respect to the output features of the penultimate layer. In CrossQ + WN we use BN to normalize the intermediate features in the entire network and we normalize the network parameters with WN. We write that we ensure the "weights remain unit norm after each gradient step by projecting them onto the unit ball”, i.e. Unit Sphere Normalization. In the future, it would of course be very interesting to investigate using Unit Sphere Normalization instead of BN on the features.
> We will update the paper to make these details much more clear to the reader.
> - **What is the relative relevance of the CrossQ part.** In our response to reviewer `7a29` , we have added a SAC + LN + WN ablation. As SAC + BN does not work (finding of the CrossQ authors) and LN provides the same scale invariance, this is an interesting comparison. We refer the reviewer to our above answer to reviewer `7a29`.
> - **Do additional resets give CrossQ + WN any benefits.** In our rebuttal to reviewer `7a29` we ran additional scaling experiments with increasing UTDs and CrossQ + WN + Resets on the Dog&Humanoid as well as the Myo Suite hard tasks (https://imgur.com/a/QPVuFfT). The only slight improvement is for Myo at UTD=5. How beneficial additional resets can be requires further experimentation which we will strongly consider for the camera ready version. However, the main take away is that ***Resetting is not required*** to train stably at higher UTD ratios with CrossQ + WN on the studied task suites.
> - **Comparison to Model-based (TD-MPC2).** In the above plot we provide TD-MPC2 results (based on the official data provided by the authors). We can see, that CrossQ + WN outperforms TD-MPC2 on the provided tasks. However, a full and in depth comparison to more model-based approaches is an interesting future direction.

---

> > ### Comment · Reviewer_9s2q · 2025-04-04
> >
> > Thanks for the additional clarifications. I'm happy to increase my score

---

### Official Review · Reviewer_7a29 · 2025-03-11

**Overall Recommendation:** 3

**Summary:**

The paper studies the scaling property of a previously proposed RL method, CrossQ, with high update-to-data ratio. CrossQ does not use target network updates and is known to be brittle to tune as also shown by the authors. The paper proposes to stabilize the training dynamics of CrossQ using weight normalization (in addition to the batch normalization that is already deployed in CrossQ) and adding back the target network updates. On the DeepMind Control Suite and Myosuite benchmark, the proposed CrossQ+WN is able to match BRO with 90% fewer parameters and does not need to periodically perform network resets.

**Claims And Evidence:**

I mainly find the following claim to be problematic: "Our proposed approach reliably scales with increasing UTD ratios".
- Figure 4 fails to show the performance scaling with increasing UTD ratios (see "experimental designs or analyses")
- The authors only test the proposed method on three UTDs (UTD=1, UTD=2, UTD=5). UTD=5 is not a very high update-to-ratio value. I find it hard to be convinced that the proposed approach can scale to higher UTDs without further evidence.

Since this claim is central to the contribution of the paper, it is a critical weakness of the paper.

**Essential References Not Discussed:**

No.

**Experimental Designs Or Analyses:**

Figure 4 – “The sample efficiency scales reliably with increasing UTD ratios.” – in the graph three UTD ratios (1, 2, and 5) perform very similarly (with overlapping confidence intervals). It is unclear if the stated conclusion can be drawn from the plot itself.

The rest of the experimental designs and analyses are all sound and valid.

**Methods And Evaluation Criteria:**

High update-to-data ratio is known to cause RL training instability. Normalization and regularization are well-known to be helpful in stabilizing RL and especially online RL training dynamics. While it is not surprising that the proposed weight normalization improves RL training stability, the proposed methods make sense and the benchmarks are challenging enough to showcase the effectiveness of the approach.

**Other Comments Or Suggestions:**

N/A

**Other Strengths And Weaknesses:**

Strengths:
- The idea is simple and the experimental results are convincing.
- The paper is easy to follow and the writing is clear.

Weaknesses:
- The paper lacks depth (especially since the method is very simple). More experiments on how weight normalization interacts with different network sizes, network architectures, initialization and target updates would be much more helpful for RL practitioners (e.g., know when WN is helpful).
- There are many other regularization techniques such as dropout, weight decay, layer normalization that could also be interesting to study. How do they interact with CrossQ?
- UTD=5 is not a very high update-to-ratio value. I find it hard to be convinced that the proposed approach can scale to higher UTDs without empirical evidence. For example, it would be helpful to show the experiments on higher UTDs (UTD>20), which is what has been studied in many prior works such as RedQ (UTD=20), DroQ (UTD=20), and prior works that use resets (UTD=32).

*References for papers that study high UTDs:*
- [RedQ] Chen, Xinyue, et al. "Randomized ensembled double q-learning: Learning fast without a model." arXiv preprint arXiv:2101.05982 (2021).
- [DroQ] Hiraoka, Takuya, et al. "Dropout q-functions for doubly efficient reinforcement learning." arXiv preprint arXiv:2110.02034 (2021).
- [Resets I] Nikishin, Evgenii, et al. "The primacy bias in deep reinforcement learning." International conference on machine learning. PMLR, 2022.
- [Resets II] D'Oro, Pierluca, et al. "Sample-efficient reinforcement learning by breaking the replay ratio barrier." Deep Reinforcement Learning Workshop NeurIPS 2022. 2022.

**Questions For Authors:**

N/A

**Relation To Broader Scientific Literature:**

The paper presents a practical and simple method to improve the stability of CrossQ. In the field of sample efficient online RL, how to properly scale RL agents with high update-to-data ratio is an open question and one of the most commonly used tricks is to periodically reset the network to restore the plasticity of the agent. It is an effective yet unsatisfying approach as it forces the RL agent to unlearn before it relearns it better. The paper takes a step towards developing simple RL algorithms that can scale with the update-to-data ratio without the need of periodic resets.

**Theoretical Claims:**

There is no new theory in the paper. Both theorems in the paper are from prior work.

---

> ### Author Rebuttal · Authors · 2025-04-01
>
> We thank the reviewer for their thorough and extensive review. We were happy to read that they found our experimental results convincing. We hope that our rebuttal manages to address their remaining concerns and open questions.  We ran many additional experiments and ablations for the rebuttal, for which we provide anonymized URLs below.
> ****
>
> **Scaling Claim.** To investigate the scaling behavior of CrossQ + WN, we ran additional experiments for UTD={10,20} and present the scaling behavior in Figure (https://imgur.com/a/QPVuFfT). In this Figure, we focus on the final performance at 1M steps, which is common practice among the baselines.
>
> We observe three things:
>
> 1. CrossQ + WN **scales with increasing UTD** ratios, and plateaus on the provided tasks after UTD > 5. In general, we argue that the fact that the method plateaus at some point could be expected.
> 2. CrossQ + WN **remains stable for larger UTD ratios**, i.e. there are no performance drops (like the ones vanilla CrossQ suffers from).
> 3. CrossQ + WN **performance is competitive** with all other baselines that we provide here.  Especially on the challening Myo Hard tasks. This includes approaches with much larger network architecutes (BRO, Simba), model-based approaches (TD-MPC2) and resetting (BRO, CrossQ + WN + Reset). CrossQ + WN is a relatively simple algorithm in comparison. The integration of other implementation details and ideas from the baselines could be interesting for future work to scale even further.
>
> We believe, under those aspects, the phrasing “reliably scales with increasing UTD ratios” is justified. However, if the reviewers still disagree, we are open to suggestions for different phrases.
>
> **Depth of the Analysis.** We agree with the reviewer's point and appreciate the many suggestions. In the limited rebuttal time, we have attempted to perform as many additional ablations and investigations as we could. We focused on
>
> - higher UTDs and scaling behavior
> - Metrics and analysis on the effectiveness of WN and why/how WN improves learning (see response to reviewer WbYM for details and Figures (https://imgur.com/a/o1ngeJD) and (https://imgur.com/a/bOnagfB))
> - Ablations on **weight decay** and **layer normalization**. Figure (https://imgur.com/a/SRgUd0D) shows additional ablations, where we used L2 regularization with varying coefficients instead of WN (please refer to our answer to reviewer `WbYM`, where we analyze the results in detail). We can see that while L2 regularization can improve the performance of vanilla CrossQ, it requires tuning the penalty coefficient per task suite, and overall performance is significantly worse than CrossQ + WN.
> While this deserves more investigation, we believe that WN is the more elegant solution since it does not require tuning and does not introduce additional hyperparameters. Moreover, we hypothesize that the constant parameter norm achieved through WN is more desirable.
> To investigate LN, we ran SAC + LN + WN experiments (since CrossQ does not work with LN according to the original authors). We find that SAC + LN + WN does not manage to reach the performance of CrossQ + WN.
> In the future, we plan to run additional ablations and experiments on network architectures, target updates and initialisations as suggested.

---

> > ### Comment · Reviewer_7a29 · 2025-04-05
> >
> > Thanks for the additional experiments. The new results convinced me the effectiveness of CrossQ+WN and made the paper stronger. I will increase my score from 2 to 3.

---

### Official Review · Reviewer_WbYM · 2025-03-11

**Overall Recommendation:** 3

**Summary:**

This paper enhances the sample efficiency of reinforcement learning (RL) by improving CrossQ, a model-free algorithm that leverages Batch Normalization (BN). While CrossQ excelled at low update-to-data (UTD) ratios, it struggled to scale reliably. The authors found that scaling the UTD ratio in CrossQ leads to a sharp increase in weight norm, which destabilizes training. To address this, they incorporate Weight Normalization (WN), which mitigates this effect, stabilizes training and enables effective scaling to higher UTD values. Experiments on DeepMind Control and MyoSuite show that CrossQ + WN significantly improves stability and sample efficiency, providing a simple yet effective enhancement to state-of-the-art RL methods.

**Claims And Evidence:**

The paper claims that Weight Normalization (WN) stabilizes CrossQ at high update-to-data (UTD) ratios, but the evidence is incomplete:

1. Stabilization of Weight Norm: The authors do not show whether WN effectively regulates weight norms across layers. Tracking weight norm dynamics throughout training would strengthen this claim.
2. Why WN Improves Performance: The authors attribute performance gains to a more stable effective learning rate, referencing prior work by Lyle. However, they do not explicitly demonstrate that combining WN and Batch Normalization (BN) results in a consistent effective learning rate. Further, other plasticity measures, such as dormant neurons or feature rank, are not discussed. These aspects could provide a more comprehensive explanation of why WN improves CrossQ's performance.

**Essential References Not Discussed:**

The paper lacks a discussion of key works on scaling and sample efficiency in deep RL.

1. Data-Efficient Reinforcement Learning with Self-Predictive Representations, ICLR’21.
2. Towards Deeper Deep Reinforcement Learning with Spectral Normalization, NeurIPS’21.
3. Mastering Diverse Domains through World Models, arXiv’23.
4. Bigger, Better, Faster: Human-level Atari with human-level efficiency, ICML’23.
5. PLASTIC: Improving Input and Label Plasticity for Sample Efficient Reinforcement Learning, NeurIPS’23.
6. TD-MPC2:Scalable, Robust World Models for Continuous Control, ICLR’24.
7. Mixtures of Experts Unlock Parameter Scaling for Deep RL, ICML’24.
8. Overestimation, Overfitting, and Plasticity in Actor-Critic: the Bitter Lesson of Reinforcement Learning, ICML’24.
9. SimBa: Simplicity Bias for Scaling Up Parameters in Deep Reinforcement Learning, ICLR’25.
10. Towards General-Purpose Model-Free Reinforcement Learning, ICLR’25.
11. MAD-TD: Model-Augmented Data stabilizes High Update Ratio RL, ICLR’25.

**Experimental Designs Or Analyses:**

1. Unclear Mechanism Behind Performance Gains: The authors attribute improvements to a more stable effective learning rate, citing Lyle, but do not demonstrate that combining WN and Batch Normalization (BN) ensures this stability. Other plasticity measures, such as dormant neurons or feature rank, are also unexplored, leaving the underlying mechanism unclear.
2. Visualization Issues: The current visualizations make it difficult to compare methods. A bar chart for each domain would improve clarity when comparing performance against baselines.
3. Limited Baselines: The paper does not compare CrossQ + WN to other relevant baselines like TD-MPC2, SimBa, or Mr.Q. While this is a minor concern, including at least one of these baselines—if computationally feasible—would strengthen the evaluation.

**Methods And Evaluation Criteria:**

1. The rationale for using Weight Normalization (WN) over standard L2 regularization is not well justified. While the authors argue that increasing weight norms can be harmful, they do not explain why L2 regularization alone is insufficient across layers. Additionally, the decision to project the layer before Batch Normalization (BN) rather than applying L2 regularization directly lacks clear motivation. A comparison of CrossQ + WN against varying strengths of L2 regularization would help clarify whether WN offers a distinct advantage.
2. Furthermore, while the authors evaluate CrossQ + WN on DeepMind Control (DMC) and MyoSuite, they do not compare it to CrossQ in MuJoCo, where the original CrossQ paper demonstrated strong performance. Including this comparison would provide a more complete assessment of WN’s impact across different environments.

**Other Comments Or Suggestions:**

I appreciate the idea and direction, but the paper needs more completeness. Strengthening connections to related work, providing a deeper analysis of Weight Normalization’s effects, and offering a more thorough comparison with existing methods would improve its clarity and impact. I am open to increasing my score if these concerns are well addressed.

**Other Strengths And Weaknesses:**

n/a

**Questions For Authors:**

n/a

**Relation To Broader Scientific Literature:**

Improving sample efficiency is a key concern in robotics, where data collection is costly and constrained by real-world limitations.

**Theoretical Claims:**

I've checked a proof of invariance in the Appendix.

---

> ### Author Rebuttal · Authors · 2025-04-01
>
> We thank the reviewer for their thorough and extensive review. We were happy to read that they appreciate the idea and direction of our work and are open to increasing the score. We hope that our rebuttal manages to address their concerns and open questions.  We ran many additional experiments and ablations for the rebuttal, for which we provide anonymized URLs below.
>
> ****
> **Deeper Analysis of Weight Normalization’s Effects.** Our rebuttal focuses on showing scaling behavior, the effective regularization of weight norms, additional plasticity metrics, and ablations with L2 regularizations.
>
> The results are aggregated per domain and over 10 seeds per environment. Due to restricted computing resources and the limited rebuttal period, we focused on the hard tasks, which total 12 environments (Dog&Humanoid for DMC and the “-hard” environments for Myo Suite).
>
> - **Effectiveness across layers.** Figure (https://imgur.com/a/o1ngeJD) shows per-layer parameter norms during training. We can see the effectiveness of CrossQ + WN in keeping parameter norms on the intermediate layers (0,1) constant and reasonable parameter norms on the output layer (2) (which is not normalized).
> On the contrary, vanilla CrossQ shows growing parameter norms on the intermediate layers. This is strongly emphasized by increasing the UTD.
> - **Why WN Improves Performance.** The following Figure (https://imgur.com/a/bOnagfB) shows, the suggested metrics to measure plasticity. We can see, that feature and parameters norms grow drastically for vanilla CrossQ. The addition of WN mitigates that growth entirely. Due to the squared Bellmann error objective, the gradient norms are (as to be expected) visibly correlated with the Q values. The ELR remains on a constant, suitable level for CrossQ + WN and stable across training. For vanilla CrossQ, the ELR is significantly smaller and near zero (which supports our hypothesis), which can be linked to the much larger and growing parameter norms. Dead / Dormant Neurons are not an issue in either case, with less that 2% and  less than 0.5% respectively.
> - **"Why WN projection before BN?”**
> WN is not part of the network architecture, i.e. not part of the forward pass. As such, the projection does not happen “before BN”. Rather, after each gradient step, we rescale the weights (WN) to unit norm. Theorems 4.1 and 4.2, justify this, since (thanks to BN) the network is scale invariant with respect to the parameter norms. While the rescaling does not influence the predictions, the benefit of such a rescaling operation is that the ELR is kept constant (since the magnitude of the weights stays constant).
> - **Why WN over L2?** L2 regularization penalizes growing weights and introduces a bias towards a smaller ELR. However, attempting to control the weight magnitude through an L2 penalty term does not guarantee a certain weight magnitude and usually requires tuning (maybe even scheduling over time). To investigate, we ran additional ablations, where we used L2 instead of WN to confirm the above hypotheses in Figure (https://imgur.com/a/SRgUd0D).
> We can see that while L2 regularization can improve performance in combination with CrossQ, tuning the penalty coefficient per task suite is required, however. Overall performance is significantly worse than CrossQ + WN.
> While this deserves more investigation, we believe that WN is the more elegant solution since it does not require tuning and does not introduce additional hyperparameters. Moreover, we hypothesize that the constant parameter norm achieved by WN is more desirable.
>
> **Visual Clarity.** We improve the visual clarity of our results by providing per-domain bar charts, as suggested (https://imgur.com/a/p4BZVVP). For comparison, we have included results for TD-MPC2 and Simba (we used the official results provided by the authors).
>
> **Strengthen connections to related work.** We thank the reviewer for the list of additional references. We will make sure to discuss all the provided references in order to position our work better within the related literature.

---

### Official Review · Reviewer_Cny1 · 2025-03-13

**Overall Recommendation:** 2

**Summary:**

The paper proposes an enhancement to the CrossQ reinforcement learning framework by integrating weight normalization (WN) with the existing batch normalization (BN) approach. The primary goal is to stabilize training when using higher update-to-data (UTD) ratios, which are typically associated with improved sample efficiency but can lead to training instabilities. The algorithm reintroduces target networks, which were removed from CrossQ and was considered as a positive effect. The paper demonstrates that by controlling the growth of network weight norms through WN, the modified algorithm---referred to as CrossQ + WN---can scale with increasing UTD ratios. The paper presents empirical results on DeepMind Control Suite and Myosuite benchmarks, comparing against baselines.

**Claims And Evidence:**

Claim: Integrating weight normalization (WN) into CrossQ improves training stability and scalability with higher UTD ratios.
Evidence: Experiments and ablation studies on DeepMind Control Suite and Myosuite benchmarks show that WN controls network weight norm growth and enables stable learning across varying UTD ratios.

**Essential References Not Discussed:**

Elsayed, M., Vasan, G., & Mahmood, A. R. (2024). Streaming deep reinforcement learning finally works. arXiv preprint arXiv:2410.14606.

Vasan, G., Elsayed, M., Azimi, S. A., He, J., Shahriar, F., Bellinger, C., White, M., & Mahmood, A. R. (2024). Deep policy gradient methods without batch updates, target networks, or replay buffers. In The Thirty-eighth Annual Conference on Neural Information Processing Systems (NeurIPS).

The above works show that you can make continuous control work without using a replay buffer and introduced similar normalization, the paper doesn't compare to the above method.

**Experimental Designs Or Analyses:**

Mentioned above in Methods and Evaluation Criteria.

**Methods And Evaluation Criteria:**

The methods and evaluation criteria are largely appropriate for continuous control tasks. However, the overlapping confidence intervals make it difficult to draw clear conclusions about the benefits of increased UTD ratios or the addition of WN. Ideally, the study would benefit from either running more random seeds to achieve tighter confidence intervals or designing controlled experiments that isolate the effects of these modifications. Additionally, expanding the evaluation to include discrete control tasks, vision-based tasks, or environments with inherent stochasticity would provide a more comprehensive assessment of the method’s applicability and robustness.

**Other Comments Or Suggestions:**

Line 106 (left col): $\pi^\star = \arg\max_\pi \mathbb{E}_{s\sim\mu_0} [ \sum_a \pi(s,a) Q^\pi(s,a)]$.

**Other Strengths And Weaknesses:**

Strength: The paper is simple and easy to read.
Weaknesses: CrossQ is an algorithm and the paper seems to be improving on it, rather than talking about simply improving algorithms for contrinous control, there exists methods which implements similar heuristics and have not been compared against, referenced below, comparing performance against these would be essential to understand if most of the normalization provided in them are simply enough to offer good sample efficiency.

Also, better empirical evaluation is required to support the claims made in the paper.

**References:**

Elsayed, M., Vasan, G., & Mahmood, A. R. (2024). Streaming deep reinforcement learning finally works. arXiv preprint arXiv:2410.14606.

Vasan, G., Elsayed, M., Azimi, S. A., He, J., Shahriar, F., Bellinger, C., White, M., & Mahmood, A. R. (2024). Deep policy gradient methods without batch updates, target networks, or replay buffers. In The Thirty-eighth Annual Conference on Neural Information Processing Systems (NeurIPS).

**Questions For Authors:**

I have two concerns:
1. Number of random seeds for evaluation because of overlapping CIs.
2. Comparison to existing streaming RL methods.

**Relation To Broader Scientific Literature:**

There exists paper and methods which don't rely on reusing data and still perform as well as the state of the art. The work fits into the broader domain of continuous control tasks

Elsayed, M., Vasan, G., & Mahmood, A. R. (2024). Streaming deep reinforcement learning finally works. arXiv preprint arXiv:2410.14606.

Vasan, G., Elsayed, M., Azimi, S. A., He, J., Shahriar, F., Bellinger, C., White, M., & Mahmood, A. R. (2024). Deep policy gradient methods without batch updates, target networks, or replay buffers. In The Thirty-eighth Annual Conference on Neural Information Processing Systems (NeurIPS).

**Theoretical Claims:**

None

---

> ### Author Rebuttal · Authors · 2025-04-01
>
> We thank the reviewer for their thorough review and questions. We were especially pleased that they acknowledged the appropriateness of our methods and evaluation criteria and that they found the paper easy to read.
>
> In the following, we want to address the reviewer’s three main concerns individually: empirical evaluation, comparison to streaming RL methods, and the extension to other domains. We ran many additional experiments and ablations for the rebuttal, for which we provide anonymized URLs below.
> ****
> **Empirical Evaluation.**
> For CrossQ + WN, Figure 1. shows a clear improvement from UTD 1 to 5 on all task suites. The improvement gap and confidence intervals are approximately the same as between BRO UTD=2 and UTD=5. More importantly, the improvement over vanilla CrossQ is significant, and the overall performance is competitive to BRO (while CrossQ + WN is algorithmically much simpler with 90% fewer parameters).
>
> We believe that our empirical results are quite strong and extensive. We present experiments across 25 continuous control tasks with 10 random seeds each, which aligns with BRO and Simba. While we agree that more seeds can reduce evaluation noise, we believe this setting is a good tradeoff for the required compute resources. However, we will increase the number of random seeds for the camera-ready version, should the reviewer insist.
>
> We agree that Figure 4. can be visually improved and does not bring the point of scaling across that well. To that end, we have run experiments with higher UTD ratios (10,20) and created a new Figure (https://imgur.com/a/QPVuFfT) that presents the scaling behaviors with respect to final performance (at 1M timesteps), as is common practice in other works (BRO, Simba). It shows
>
> - a substantial improvement over vanilla CrossQ (which shows dropping performance with higher UTDs)
> - CrossQ + WN is competitive to all baselines for varying UTDs, scales initially and stably plateaus on the provided tasks.
> - Adding periodic resetting can slightly help in Myo Hard, but does not help on DMC. Overall, this can be investigated more. However, the main takeaway is that CrossQ + WN does not require resetting in order to scale stably.
>
> > the overlapping confidence intervals make it difficult to draw clear conclusions about the benefits […] the addition of WN
> >
>
> We disagree with this statement and point the reviewer to Figure 1, which clearly shows that the addition of WN enables CrossQ to scale. Without WN, CrossQ shows worse performance with increasing UTDs. The benefits of higher UTDs are especially pronounced on the Hard tasks (Dog&Humaoid, Myo Hard), where the baselines take more time to learn.
>
> ****
> **Comparison to existing streaming RL methods.**
> We thank the reviewer for the references and believe it is interesting to discuss the relation to these works in the related work section. As such, we will integrate them into our discussion. We view these references as *concurrent work* in accordance with the ICML reviewing guidelines.
> Further, while the removal of replay buffers is interesting, it is out of the scope for this work where we focus on sample efficiency. The provided streaming RL approaches use on the order of 10M samples to learning the dog environments (compared to <1M for the algorithms we consider in this work).
> ****
>
> **Extension to different tasks.**
> We agree that the extension to discrete control tasks and vision-based tasks would be an interesting study. Given the number of experiments and evaluations in our current draft, and the variaous experiments we added during the rebuttal, we believe this deserves a separate study and would leave this investigation for future work.
>
> We want to note that while we do not provide experiments with stochastic dynamics yet, the Myo Suite hard tasks do have inherent stochasticity. In these tasks, goals and initial conditions are randomized.

---

> > ### Comment · Reviewer_Cny1 · 2025-04-03
> >
> > ### Motivation for the Paper:
> > The title Scaling Off-Policy Reinforcement Learning with Batch and Weight Normalization suggests a general scalability improvement across off-policy RL methods. However, the paper only investigates the effect of weight normalization on the specific algorithm, CrossQ. I believe the title and framing should more accurately reflect that the work demonstrates an improvement over a particular algorithm rather than implying broad scalability across diverse methods. Typically, "scaling" implies the ability to handle more complex problems with increased compute/time, which is not clearly demonstrated here.
> >
> > ### Empirical Evaluation:
> >
> > **Improvement of CrossQ + WN**:
> > The goal is to provide a better algorithm, but Figure 1 shows significant overlap between BRO and CrossQ. The choice of UTD ratios from 1 to 5 appears too narrow, and the presented new figures---even when suggesting that BRO might be more scalable with higher UTD---lack sufficient error bars.
> >
> > **Statistical Robustness**:
> > As an empirical study, this work would benefit from proper statistical analyses with an increased number of seeds. I would be happy to see more seeds included, but I cannot improve my score until I see such improvements. In Figure 4, which examines scaling with respect to UTD, the overlapping standard errors across all UTD values make it unclear if the method truly improves performance.
> >
> > While I am open to not testing on additional tasks (and understand concurrence), I remain unconvinced by the current set of results without stronger statistical validation. If compute constraints are an issue, performing these ablations on simpler environments might help clarify the benefits of the approach.
> >
> > I hope these comments help refine the paper

---

> > > ### Author Response · Authors · 2025-04-07
> > >
> > > We thank the reviewer for their comments and want to provide some additional comments and clarifications.
> > >
> > > > the *overlapping standard errors* […].
> > > >
> > >
> > > There seems to be a misunderstanding. To clarify, we *DO NOT show the standard error* for the confidence intervals. Instead, we present **95% stratified bootstrap confidence intervals** with 50.000 resampling iterations.
> > >
> > > > As an empirical study, this work would benefit from ***proper statistical analyses*** […].
> > > >
> > >
> > > Our evaluation protocol follows the suggestions of Agarwal et al. 2022 (Deep Reinforcement Learning at the Edge of the Statistical Precipice, NeurIPS). This protocol is statistically well-motivated. The presented inter quartile mean (IQM) is both robust to outliers (unlike the mean) and more statistically efficient than the median. Stratified bootstrap confidence intervals take statistical uncertainty into account. Agarwal et al. find that $N = 10$ runs already provide good CIs for the IQM. The results we present are aggregated over a total of 250 seeds ((15 DMC envs + 10 Myo Envs) * 10 seeds each).
> > >
> > > To convince the reviewer, we re-plotted our results (https://imgur.com/a/engjRM6). For visual clarity, we have
> > >
> > > - removed UTD=2 to observe the gap between UTD=1 and 5 better.
> > > - changed the colors for better visual separability
> > >
> > > We plot **IQM** and **95% stratified bootstrap confidence intervals**.
> > >
> > > - On the very right, we aggregate over all 15 DMC envs (150 seeds) until 1M steps. We observe that UTD=5 dominates UTD=1. CIs overlap in some parts, however, not everywhere and keep a margin to the IQM.
> > > - In the middle, we aggregate over the 8 easy and medium DMC envs (80 seeds). Since these environments are learned faster, and the improvement of the higher UTD happens in the earlier stage of training, we concentrate on the first 500k steps for this plot. We observe a large separation between the IQMs and CIs of the different UTDs.
> > > - On the left, we aggregate over the 7 hard (Dog & Humanoid) DMC envs (70 seeds). These environments are harder to learn, and the benefit of UTD=5 shows later in training. The CIs are larger and slightly overlap. However, later in training, IQM distance increases and CI overlapping decreases.
> > >
> > > We believe that our results provide **a proper statistical analysis** based on the protocol of Agarwal et al. We further believe that the benefits of larger UTDs are clearly visible, and we hope that with our new, improved visualization, we were able to convince the reviewer of the scalability of our approach.
> > >
> > > > The goal is to provide a better algorithm, but Figure 1 shows significant overlap between BRO and CrossQ.
> > > >
> > >
> > > The goal is to scale CrossQ to higher UTD ratios. We analyze why it does not scale and propose a fix. We find that we end up with an algorithm that is competitive with the current SOTA methods while being much simpler.
> > >
> > > Further, we can argue that our proposed algorithm is indeed **better** than BRO in the sense that the sample efficiency is competitive and at the same time the algorithm is simpler and more lightweight: 90% smaller networks, no distributional critics, does not require periodic resetting, and no dual policy optimistic exploration needed.
> > >
> > > We hope that we could clarify the open questions and want to thank the reviewer again for their time and their review.

---

### Decision · Program_Chairs · 2025-05-01

**Decision:**

Reject

**Comment:**

This work investigates the impact of weight normalization (WN) when integrated into the CrossQ method. Specifically, the authors demonstrate that, alongside batch normalization, incorporating WN helps stabilize training at higher update-to-data (UTD) ratios. Higher UTD ratios generally enhance sample efficiency but can induce training instabilities, which the addition of WN appears to mitigate. Reviewers agreed that this work presents interesting and valuable observations, particularly regarding the role of WN within CrossQ, despite the simplicity of the proposed method.

However, several major concerns remain, and addressing these will significantly improve the paper:

* Currently, the paper frames the results as broadly applicable. However, the empirical evidence presented pertains specifically to CrossQ. The authors should provide evidence supporting broader applicability and scalability.

* The study's empirical evaluation is currently limited. It would greatly benefit from expanded experiments, exploring how WN interacts with diverse network architectures, varying network sizes, initialization methods, and different target update strategies. Additionally, overlapping confidence intervals make it difficult to draw reliable conclusions about performance improvements.

During the discussion phase, some concerns were partially addressed. However, the paper still requires revision to incorporate the new results and clarifications provided in the authors’ responses.